# Bellman Error Based Feature Generation using Random Projections on Sparse Spaces

**Mahdi Milani Fard, Yuri Grinberg, Amir massoud Farahmand, Joelle Pineau, Doina Precup**
School of Computer Science
McGill University
Montreal, Canada
{mmilan1,ygrinb,amirf,jpineau,dprecup}@cs.mcgill.ca

## Abstract

This paper addresses the problem of automatic generation of features for value function approximation in reinforcement learning. Bellman Error Basis Functions (BEBFs) have been shown to improve policy evaluation, with a convergence rate similar to that of value iteration. We propose a simple, fast and robust algorithm based on random projections, which generates BEBFs for sparse feature spaces. We provide a finite sample analysis of the proposed method, and prove that projections logarithmic in the dimension of the original space guarantee a contraction in the error. Empirical results demonstrate the strength of this method in domains in which choosing a good state representation is challenging.

## 1 Introduction

Policy evaluation, i.e. computing the expected return of a given policy, is at the core of many reinforcement learning (RL) algorithms. In large problems, it is necessary to use function approximation in order to perform this task; a standard choice is to hand-craft parametric function approximators, such as a tile coding, radial basis functions or neural networks. The accuracy of parametrized policy evaluation depends crucially on the quality of the features used in the function approximator, and thus often a lot of time and effort is spent on this step. The desire to make this process more automatic has led to a lot of recent work on feature generation and feature selection in RL (e.g. [1, 2, 3, 4, 5]).

An approach that offers good theoretical guarantees is to generate features in the direction of the Bellman error of the current value estimates (Bellman Error Based features, or BEBF). Successively adding exact BEBFs has been shown to reduce the error of a linear value function estimator at a rate similar to value iteration, which is the best one could hope to achieve [6]. Unlike fitted value iteration [7], which works with a fixed feature set, iterative BEBF generation gradually increases the complexity of the hypothesis space by adding new features and thus does not diverge, as long as the error in the generation does not cancel out the contraction effect of the Bellman operator [6]. Several successful methods have been proposed for generating features related to the Bellman error [5, 1, 4, 6, 3]. In practice however, these methods can be computationally expensive when applied in high dimensional input spaces.

With the emergence of more high-dimensional RL problems, it has become necessary to design and adapt BEBF-based methods to be more scalable and computationally efficient. In this paper, we present an algorithm that uses the idea of applying random projections specifically in very large and sparse feature spaces (e.g. $10^5 - 10^6$ dimensions). The idea is to iteratively project the original features into exponentially lower-dimensional spaces. Then, we apply linear regression in the smaller spaces, using temporal difference errors as targets, in order to approximate BEBFs.

Random projections have been studied extensively in signal processing [8, 9] as well as machine learning [10, 11, 12, 13]. In reinforcement learning, Ghavamzadeh et al. [14] have used random projections in conjunction with LSTD and have shown that this can reduce the estimation error,

at the cost of a controlled bias. Instead of compressing the feature space for LSTD, we focus on the BEBF generation setting, which offers better scalability and more flexibility in practice. Our algorithm is well suited for sparse feature spaces, naturally occurring in domains with audio and video inputs [15], and also in tile-coded and discretized spaces.

We carry out a finite sample analysis, which helps determine the sizes that should be used for the projections. Our analysis holds for both finite and continuous state spaces and is easy to apply with discretized or tile-coded features, which are popular in many RL applications. The proposed method compares favourably, from a computational point of view, to many other feature extraction methods in high dimensional spaces, as each iteration takes only poly-logarithmic time in the number of dimensions. The method provides guarantees on the reduction of the error, yet needs minimal domain knowledge, as we use agnostic random projections.

Our empirical analysis indicates that the proposed method provides similar results to $L^2$-regularized LSTD, but scales much better in time complexity as the observed sparsity decreases. It significantly outperforms $L^1$-regularized methods both in performance and computation time. The algorithm seems robust to the choice of parameters and has small computational and memory complexity.

## 2 Notation and Background

Throughout this paper, column vectors are represented by lower case bold letters, and matrices are represented by bold capital letters. $|.|$ denotes the size of a set, and $\mathcal{M}(\mathcal{X})$ is the set of measures on $\mathcal{X}$. $\|.\|_0$ is Donoho's zero "norm" indicating the number of non-zero elements in a vector. $\|.\|$ denotes the $L^2$ norm for vectors and the operator norm for matrices: $\|\mathbf{M}\| = \sup_{\mathbf{v}} \|\mathbf{M}\mathbf{v}\|/\|\mathbf{v}\|$. The Frobenius norm of a matrix is then defined as: $\|\mathbf{M}\|_F = \sqrt{\sum_{i,j} \mathbf{M}_{i,j}^2}$. Also, we denote the Moore-Penrose pseudo-inverse of a matrix $\mathbf{M}$ with $\mathbf{M}^\dagger$. The weighted $L^2$ norm of a function is defined as $\|f(\mathbf{x})\|_{\rho(\mathbf{x})} = \sqrt{\int |f(\mathbf{x})|^2 \, d\rho(\mathbf{x})}$. We focus on spaces that are large, bounded and $k$-sparse. Our state is represented by a vector $\mathbf{x} \in \mathcal{X}$ of $D$ features, having $\|\mathbf{x}\| \leq 1$. We assume that $\mathbf{x}$ is $k$-sparse in some known or unknown basis $\boldsymbol{\Psi}$: $\mathcal{X} \triangleq \{\boldsymbol{\Psi}\mathbf{z}, \text{ s.t. } \|\mathbf{z}\|_0 \leq k \text{ and } \|\mathbf{z}\| \leq 1\}$. Such spaces occur both naturally (e.g. image, audio and video signals [15]) as well as from most discretization-based methods (e.g., tile-coding).

### 2.1 Markov Decision Process

A *Markov Decision Process* (MDP) $M = (\mathcal{S}, \mathcal{A}, T, R)$ is defined by a (possibly infinite) set of states $\mathcal{S}$, a set of actions $\mathcal{A}$, a transition kernel $T : \mathcal{S} \times \mathcal{A} \rightarrow \mathcal{M}(\mathcal{S})$, where $T(.|s, a)$ defines the distribution of next state given that action $a$ is taken in state $s$, and a (possibly stochastic) bounded reward function $R : \mathcal{S} \times \mathcal{A} \rightarrow \mathcal{M}([0, R_{\max}])$. We assume discounted-reward MDPs, with the discount factor denoted by $\gamma \in [0, 1)$. At each discrete time step, the RL agent chooses an action and receives a reward. The environment then changes to a new state, according to the transition kernel.

A *policy* is a (possibly stochastic) function from states to actions. The *value of a state* $s$ for policy $\pi$, denoted by $V^\pi(s)$, is the expected value of the discounted sum of rewards ($\sum_t \gamma^t r_t$) if the agent starts in state $s$ and acts according to policy $\pi$. Let $R(s, \pi(s))$ be the expected reward at state $s$ under policy $\pi$. The value function satisfies:

$$V^\pi(s) = R(s, \pi(s)) + \gamma \int V^\pi(s') T(ds'|s, \pi(s)). \tag{1}$$

Many methods have been developed for finding the value of a policy (policy evaluation) when the transition and reward functions are known. Dynamic programming methods apply iteratively the *Bellman operator* $\mathcal{T}$ to an initial guess of the value function [16]:

$$\mathcal{T}V(s) = R(s, \pi(s)) + \gamma \int V(s') T(ds'|s, \pi(s)), \tag{2}$$

When the transition and reward models are not known, one can use a finite sample set of transitions to learn an approximate value function. When the state space is very large or continuous, the value function is also approximated using a feature vector $\mathbf{x}_s$, which is a function of the state $s$. Often, this approximation is linear: $V(s) \approx \mathbf{w}^T \mathbf{x}_s$. To simplify the derivations, we use $V(\mathbf{x})$ to directly refer to the value estimate of a state with feature vector $\mathbf{x}$.

Least-squares temporal difference learning (LSTD) and its variations [17, 18, 19] are among methods that learn a value function based on a finite sample, especially when function approximation is needed. LSTD-type methods are efficient in their use of data, but can be computationally expensive, as they rely on inverting a large matrix. Using LSTD in spaces induced by random projections is a way of dealing with this problem [14]. As we show in our experiments, if the observation space is sparse, we can also use conjugate gradient descent methods to solve the regularized LSTD problem.

Stochastic gradient descent methods are alternatives to LSTD in high-dimensional state spaces, as their memory and computational complexity per time step are linear in the number of state features, while providing convergence guarantees [20]. However, online gradient-type methods typically have slow convergence rates and do not make efficient use of the data.

## 2.2 Bellman Error Based Feature Generation

In high-dimensional state spaces, direct estimation of the value function fails to provide good results when using a small number of sampled transitions. Feature selection/extraction methods have thus been used to build better approximation spaces for the value functions [1, 2, 3, 4, 5]. Among these, we focus on methods that aim to generate features in the direction of the *Bellman error* defined as:

$$e_V(.) = \mathcal{T}V(.) - V(.). \tag{3}$$

Let $S_n = ((\mathbf{x}_t, r_t)_{t=1}^n)$ be a random sample of size $n$, collected on an MDP with a fixed policy. Given an estimate $V$ of the value function, *temporal difference (TD) errors* are defined to be:

$$\delta_t = r_t + \gamma V(\mathbf{x}_{t+1}) - V(\mathbf{x}_t). \tag{4}$$

It is easy to show that the expectation of the temporal difference at $\mathbf{x}_t$ equals the Bellman error at that point [16]. TD-errors are thus proxies to estimating the Bellman error.

Using temporal differences, Menache et al. [21] introduced two algorithms to construct basis functions for linear function approximation. Keller et al. [3] applied neighbourhood component analysis as a dimensionality reduction technique to construct a low dimensional state space based on the TD-error. In their work, they iteratively add features that would help predict the Bellman error. Parr et al. [6] later showed that any BEBF extraction method with small angular error will provably tighten the approximation error of the value function estimate. Online BEBF extraction methods have also been studied in the RL literature. The *incremental Feature Dependency Discovery* (iFDD) is a fast online algorithm to extract non-linear binary features for linear function approximation [5].

We note that these algorithms, although theoretically interesting, are difficult to apply to very large state spaces or need specific domain knowledge to generate good features. The problem lies in the large estimation error when predicting BEBFs in high-dimensional state spaces. Our proposed solution leverages the use of simple random projections to alleviate this problem.

## 2.3 Random Projections and Inner Product

Random projections have been introduced in signal processing, as an efficient method for compressing very high-dimensional signals (such as images or video). It is well known that random projections of appropriate sizes preserve enough information to exactly reconstruct the original signal with high probability [22, 9]. This is because random projections are norm and distance-preserving in many classes of feature spaces.

There are several types of random projection matrices that can be used. In this work, we assume that each entry in the projection matrix $\mathbf{\Phi}^{D \times d}$ is an i.i.d. sample from a Gaussian distribution:

$$\phi_{i,j} \sim \mathcal{N}(0, 1/d). \tag{5}$$

Recently, it has been shown that random projections of appropriate sizes preserve linearity of a target function on sparse feature spaces. A bound introduced in [11] and later tightened by [23] shows that if a function is linear in a sparse space, it is almost linear in an exponentially smaller projected space. An immediate lemma based on Theorem 2 of [23] bounds the bias induced by random projections:

**Lemma 1.** *Let $\mathcal{X}$ be a $D$-dimensional $k$-sparse space and $\mathbf{\Phi}^{D \times d}$ be a random projection according to Eqn 5. Fix $\mathbf{w} \in \mathbb{R}^D$ and $1 > \xi_0 > 0$. Then, for $\epsilon_{prj}^{(\xi_0)} = \sqrt{\frac{48k}{d} \log \frac{4D}{\xi_0}}$, with probability $> 1 - \xi_0$ :*

$$\forall \mathbf{x} \in \mathcal{X} : \left| (\mathbf{\Phi}^T \mathbf{w})^T (\mathbf{\Phi}^T \mathbf{x}) - \mathbf{w}^T \mathbf{x} \right| \leq \epsilon_{prj}^{(\xi_0)} \|\mathbf{w}\| \|\mathbf{x}\|, \tag{6}$$

Hence, projections of size $\tilde{O}(k \log D)$ preserve the linearity up to an arbitrary constant. Along with the analysis of the variance of the estimators, this helps bound the prediction error of the linear fit in the compressed space.

## 3 Compressed Linear BEBFs

In this work, we propose a new method to generate BEBFs using linear regression in a small space induced by random projections. We first project the state features into a much smaller space and then regress a hyperplane to the TD-errors. For simplicity, we assume that regardless of the current estimate of the value function, the Bellman error is always linearly representable in the original feature space. This seems like a strong assumption, but is true, for example, in virtually any discretized space, and is also likely to hold in very high dimensional feature spaces[1].

Linear function approximators can be used to estimate the value of a given state. Let $V_m$ be an estimated value function described in a linear space defined by a feature set $\Psi = \{\psi_1, \ldots \psi_m\}$. Parr et al. [6] show that if we add a new BEBF $\psi_{m+1} = e_{V_m}$ to the feature set, (with mild assumptions) the approximation error on the new linear space shrinks by a factor of $\gamma$. They also show that if we can estimate the Bellman error within a constant angular error, $\cos^{-1}(\gamma)$, the error will still shrink.

Estimating the Bellman error by regressing to temporal differences in high-dimensional sparse spaces can result in large prediction error. This is due to the large estimation error of regression in high dimensional spaces (over-fitting). However, as discussed in Lemma 1, random projections were shown to exponentially reduce the dimension of a sparse feature space, only at the cost of a controlled constant bias. A variance analysis along with proper mixing conditions can also bound the estimation error due to the variance in MDP returns. The computational cost of the estimation is also much smaller when the regression is applied in the compressed space.

### 3.1 General CBEBF Algorithm

In light of these results, we propose the *Compressed Bellman Error Based Feature Generation* algorithm (CBEBF). The algorithm iteratively constructs new features using compressed linear regression to the TD-errors, and uses these features with a policy evaluation algorithm to update the estimate of the value function.

---
**Algorithm 1** Compressed Bellman Error Based Feature Generation (CBEBF)
---
**Input:** Sample trajectory $S_n = ((\mathbf{x}_t, r_t)_{t=1}^n)$, where $\mathbf{x}_t$ is the observation received at time $t$, and $r_t$ is the observed reward; Number of BEBFs: $m$; Projection size schedule: $d_1, d_2, \ldots, d_m$
**Output:** $V(.)$: estimate of the value function

Initialize $V(.)$ to be 0 for all $\mathbf{x}$.
Initialize the set of BEBFs linear weights $\Psi \leftarrow \emptyset$.
**for** $i \leftarrow 1$ **to** $m$ **do**
    Generate projection $\mathbf{\Phi}^{D \times d_i}$ according to Eqn 5.
    Calculate TD-errors: $\delta_t = r_t + \gamma V(\mathbf{x}_{t+1}) - V(\mathbf{x}_t)$.
    Apply compressed regression:
      Let $\mathbf{u}^{d_i \times 1}$ be the result of OLS regression in the compressed space,
      using $\mathbf{\Phi}^T \mathbf{x}_t$ as inputs and $\delta_t$ as outputs.
    Add $\mathbf{\Phi u}$ to $\Psi$.
    Apply policy evaluation with features $\{\hat{e}_{\mathbf{v}}(\mathbf{x}) = \mathbf{x}^T \mathbf{v} \,|\, \mathbf{v} \in \Psi\}$ to update $V(.)$.
**end for**

---

The optimal number of BEBFs and the schedule of projection sizes need to be determined and are subjects of future work. But we show in the next section that logarithmic size projections should be enough to guarantee the reduction of error in value function prediction at each step. This makes the algorithm very attractive when it comes to computational and memory complexity, as the regression at each step is only on a small projected feature space. As we discuss in our empirical analysis, the algorithm is fast and robust with respect to the selection of parameters.

## 3.2 Simplified CBEBF as Regularized Value Iteration

Note that in CBEBF, we can use any type of value function approximation to estimate the value function in each iteration. To simplify the bias–variance analysis and avoid multiple levels of regression, we present here a simplified version of the CBEBF algorithm (SCBEBF). In the simplified version, instead of storing the features in each iteration, new features are added to the value function approximator with constant weight 1. Therefore, the value estimate is simply the sum of all generated BEBFs. As compared to the general CBEBF, the simplified version trivially has lower computational complexity per iteration, as it avoids an extra level of regression based on the features. It also avoids storing the features by simply keeping the sum of all previously generated coefficients.

It is important to note that once we use linear value function approximation, the entire BEBF generation process can be viewed as a regularized value iteration algorithm. Each iteration of the algorithm is a regularized Bellman backup which is linear in the features. The coefficients of this linear backup are confined to a lower-dimensional random subspace implicitly induced by the random projection used in each iteration.

## 3.3 Finite Sample Analysis of Simplified CBEBF

This section provides a finite sample analysis of the simplified CBEBF algorithm. In order to provide such analysis, we need to have an assumption on the range of observed TD-errors. This is usually possible by assuming that the current estimate of the value function is bounded, which is easy to enforce by truncating any estimate of the value function between 0 and $V_{\max} = R_{\max}/(1 - \gamma)$.

The following theorem shows how well we can estimate the Bellman error by regression to the TD-errors in a compressed space. It highlights the bias–variance trade-off with respect to the choice of the projection size.

**Theorem 2.** *Let $\boldsymbol{\Phi}^{D \times d}$ be a random projection according to Eqn 5. Let $S_n = ((\mathbf{x}_t, r_t)_{t=1}^n)$ be a sample trajectory collected on an MDP with a fixed policy with stationary distribution $\rho$, in a $D$-dimensional $k$-sparse feature space, with $D > d \geq 10$. Let $\tau$ be the forgetting time of the chain (defined in the appendix). Fix any estimate $V$ of the value function, and the corresponding TD-errors $\delta_t$'s bounded by $\pm\delta_{\max}$. Assume that the Bellman error is linear in the features with parameter $\mathbf{w}$. With compressed OLS regression we have $\mathbf{w}_{ols}^{(\boldsymbol{\Phi})} = (\mathbf{X}\boldsymbol{\Phi})^\dagger \delta$, where $\mathbf{X}$ is the matrix containing $\mathbf{x}_t$'s and $\delta$ is the vector of TD-errors. Assume that $\mathbf{X}$ is of rank larger than $d$. For any fixed $0 < \xi < 1/4$, with probability no less than $1 - \xi$, the prediction error $\left\|\mathbf{x}^T \boldsymbol{\Phi} \mathbf{w}_{ols}^{(\boldsymbol{\Phi})} - e_V(\mathbf{x})\right\|_{\rho(\mathbf{x})}$ is bounded by:*

$$12\,\alpha\epsilon_{prj}^{(\xi/4)}\|\mathbf{w}\|\|\mathbf{x}\|_\rho\sqrt{\frac{1}{d\nu}} + 4\alpha\epsilon_{prj}^{(\xi/4)}\|\mathbf{w}\|\sqrt{\frac{d\tau}{n\nu}\log\frac{d}{\xi}} + 2\alpha\delta_{\max}\|\mathbf{x}\|_\rho\sqrt{\frac{\kappa d}{n\nu}\log\frac{d}{\xi}} \qquad (7)$$

*where $\epsilon_{prj}^{(\xi/4)}$ is according to Lemma 1, $\kappa$ and $\nu$ are the condition number and the smallest positive eigenvalue of the empirical gram matrix $\frac{1}{n}\boldsymbol{\Phi}^T\mathbf{X}^T\mathbf{X}\boldsymbol{\Phi}$, and we define maximum norm scaling factor $\alpha = \max(1, \max_{\mathbf{z}\in\mathcal{X}}\left\|\mathbf{z}^T\boldsymbol{\Phi}\right\|/\|\mathbf{z}\|)$.*

A detailed proof is included in the appendix. The sketch of the proof is as follows: Lemma 1 suggests that if the Bellman error is linear in the original features, the bias due to the projection can be bounded within a controlled constant error with logarithmic size projections. If the Markov chain uniformly quickly forgets its past, one can also bound the *on-measure* variance part of the error. The variance terms, of course, go to 0 as the number of sampled transitions $n$ goes to infinity.

Theorem 2 can be further simplified by using concentration bounds on random projections as defined in Eqn 5. The norm of $\boldsymbol{\Phi}$ can be bounded using the bounds discussed in Candès and Tao [8]; we have with probability $1 - \delta_\Phi$:

$$\|\boldsymbol{\Phi}\| \leq \sqrt{D/d} + \sqrt{(2\log(2/\delta_\Phi))/d} + 1 \quad \text{and} \quad \|\boldsymbol{\Phi}^\dagger\| \leq \left[\sqrt{D/d} - \sqrt{(2\log(2/\delta_\Phi))/d} - 1\right]^{-1}.$$

Similarly, when $n > d$, we expect the smallest and biggest singular values of $\mathbf{X}\boldsymbol{\Phi}$ to be of order of $\tilde{O}(\sqrt{n/d})$. Thus we have $\kappa = O(1)$ and $\nu = O(1/d)$. Projections are norm-preserving and thus

$\alpha \simeq 1$. Assuming that $n = \tilde{O}(d^2)$, we can rewrite the bound on the error up to logarithmic terms as:

$$\tilde{O}\left(\|\mathbf{w}\|\|\mathbf{x}\|_{\rho(\mathbf{x})}\sqrt{k\log D/d}\right) + \tilde{O}\left(d/\sqrt{n}\right). \tag{8}$$

The first term is a part of the bias due to the projection (excess approximation error). The rest is the combined variance terms that shrink with larger training sets (estimation error). We clearly observe the trade-off with respect to the compressed dimension $d$. With the assumptions discussed above, we can see that projection of size $d = \tilde{O}(k\log D)$ should be enough to guarantee arbitrarily small bias, as long as $\|\mathbf{w}\|\|\mathbf{x}\|_{\rho(\mathbf{x})}$ is small. Thus, the bound is tight enough to prove reduction in the error as new BEBFs are added to the feature set.

Note that this bound matches that of Ghavamzadeh et al. [14]. The variance term is of order $\sqrt{d/n\nu}$. Thus, the dependence on the smallest eigenvalue of the gram matrix makes the variance term order $d/\sqrt{n}$ rather than the expected $\sqrt{d/n}$. We expect the use of ridge regression instead of OLS in the inner loop of the algorithm to remove this dependence and help with the convergence rate (see appendix).

As mentioned before, our simplified version of the algorithm does not store the generated BEBFs (such that it could later apply value function approximation over them). It adds up all the features with weight 1 to approximate the value function. Therefore our analysis is different from that of Parr et al. [6]. The following lemma (simplification of results in Parr et al. [6]) provides a sufficient condition for the shrinkage of the error in the value function prediction:

**Lemma 3.** *Let $V^\pi$ be the value function of a policy $\pi$ imposing stationary measure $\rho$, and let $e_V$ be the Bellman error under policy $\pi$ for an estimate $V$. Given a BEBF $\psi$ satisfying:*

$$\|\psi(\mathbf{x}) - e_V(\mathbf{x})\|_{\rho(\mathbf{x})} \leq \epsilon \|e_V(\mathbf{x})\|_{\rho(\mathbf{x})}, \tag{9}$$

*we have that:* $\quad \|V^\pi(\mathbf{x}) - (V(\mathbf{x}) + \psi(\mathbf{x}))\|_{\rho(\mathbf{x})} \leq (\gamma + \epsilon + \epsilon\gamma)\|V^\pi(\mathbf{x}) - V(\mathbf{x})\|_{\rho(\mathbf{x})}. \tag{10}$

Theorem 2 (simplified in Equation (8)) does not state the error in terms of $\|e_V(\mathbf{x})\|_\rho = \|\mathbf{w}^T\mathbf{x}\|_\rho$, as needed by this lemma, but rather does it in terms of $\|\mathbf{w}\|\|\mathbf{x}\|_\rho$. Therefore, if there is a large gap between these terms, we cannot expect to see shrinkage in the error (we can only show that the error can be shrunk to a bounded uncontrolled constant). Ghavamzadeh et al. [14] and Maillard and Munos [10, 12] provide some discussion on the cases were $\|\mathbf{w}^T\mathbf{x}\|_\rho$ and $\|\mathbf{w}\|\|\mathbf{x}\|_\rho$ are expected to be close. These cases include when the features are rescaled orthonormal basis functions and also with specific classes of wavelet functions.

The dependence on the norm of $\mathbf{w}$ is conjectured to be tight by the compressed sensing literature [24], making this bound asymptotically the best one can hope for. This dependence also points out an interesting link between our method and $L^2$-regularized LSTD. We expect ridge regression to be favourable in cases where the norm of the weight vector is small. The upper bound on the error of compressed regression is also smaller when the norm of $\mathbf{w}$ is small.

**Lemma 4.** *Assume the conditions of Theorem 2. Further assume for some constants $c_1, c_2, c_3 \geq 1$:*

$$\|\mathbf{w}\| \leq c_1\|\mathbf{w}^T\mathbf{x}\|_\rho \quad and \quad \|\mathbf{x}\|_\rho \leq c_2\|\mathbf{w}^T\mathbf{x}\|_\rho \quad and \quad 1/\nu \leq c_3 d, \tag{11}$$

*There exist universal constants $c_4$ and $c_5$, such that for any $\gamma < \gamma_0 < 1$ and $0 < \xi < 1/4$, if:*

$$d \geq \alpha^2 c_1^2 c_2^2 c_3 c_4 \left(\frac{1+\gamma}{\gamma_0-\gamma}\right)^2 k\log\frac{D}{\xi} \quad and \quad n \geq (\tau + \alpha^2 c_2^2 c_3 \delta_{\max}^2 \kappa)c_5\left(\frac{1+\gamma}{\gamma_0-\gamma}\right)^2 d^2\log\frac{d}{\xi},$$

*then with the addition of the estimated BEBF, we have that with probability $1 - \xi$:*

$$\|V^\pi(\mathbf{x}) - (V(\mathbf{x}) + \psi(\mathbf{x}))\|_{\rho(\mathbf{x})} \leq \gamma_0\|V^\pi(\mathbf{x}) - V(\mathbf{x})\|_{\rho(\mathbf{x})}. \tag{12}$$

The proof is included in the appendix. Lemma 4 shows that with enough sampled transitions, using random projections of size $d = \tilde{O}\left((\frac{1+\gamma}{\gamma_0-\gamma})^2 k\log D\right)$ guarantees contraction in the error by a factor of $\gamma_0$. Using union bound over $m$ iterations of the algorithm, we prove that projections of size $d = \tilde{O}\left((\frac{1+\gamma}{\gamma_0-\gamma})^2 k\log(mD)\right)$ and a sample of transitions of size $n = \tilde{O}\left((\frac{1+\gamma}{\gamma_0-\gamma})^2 d^2\log(md)\right)$ suffices to shrink the error by a factor of $\gamma_0^m$ after $m$ iterations.

# 4 Empirical Analysis

We conduct a series of experiments to evaluate the performance of our algorithm and compare it against viable alternatives. Experiments are performed using a simulator that models an autonomous helicopter in the flight regime close to hover [25]. Our goal is to evaluate the value function associated with the manually tuned policy provided with the simulator. We let the helicopter free fall for 5 time-steps before the policy takes control. We then collect 100 transitions while the helicopter hovers. We run this process multiple times to collect more trajectories on the policy.

The original state space of the helicopter domain consists of 12 continuous features. 6 of these features corresponding to the velocities and position, capture most of the data needed for policy evaluation. We use tile-coding on these 6 features as follows: 8 randomly positioned grids of size $16 \times 16 \times 16$ are placed over forward, sideways and downward velocity. 8 grids of similar structure are placed on features corresponding to the hovering coordinates. The constructed feature space is thus of size 65536. Note that our choice of tile-coding for this domain is for demonstration purposes.

Since the true value function is not known in our case, we evaluate the performance of the algorithm by measuring the *normalized return prediction error* (NRPE) on a large test set. Let $U(\mathbf{x}_i)$ be the empirical return observed for $\mathbf{x}_i$ in a testing trajectory, and $\bar{U}$ be its average over the testing measure $\mu(x)$. We define $\text{NRPE}(V) = \|U(\mathbf{x}) - V(\mathbf{x})\|_{\mu(\mathbf{x})}/\|U(\mathbf{x}) - \bar{U}\|_{\mu(\mathbf{x})}$. Note that the best constant predictor has NRPE = 1.

We start by an experiment to observe the behaviour of the prediction error in SCBEBF as we run more iterations of the algorithm. We collect 3000 sample transitions for training. We experiment with 3 schedules for the projection size: (1) Fix $d = 300$ for 300 steps. (2) Fix $d = 30$ for 300 steps. (3) Let $d$ decrease with each iteration $i$: $d = \lfloor 300e^{-i/30} \rfloor$. Figure 1 (left) shows the error averaged over 5 runs. When $d$ is fixed to a large number, the prediction error drops rapidly, but then rises due to over-fitting. This problem can be mitigated by using a smaller fixed projection size at the cost of slower convergence. In our experiments, we find a gradual decreasing schedule to provide fast and robust convergence with minimal over-fitting effects.

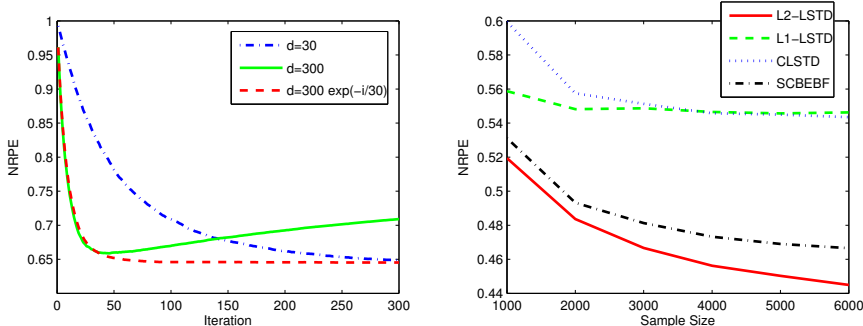

Figure 1: Left: NRPE of SCBEBF for different number of projections, under different choices of $d$, averaged over 5 runs. Right: Comparison of the prediction error of different methods for varying sample sizes. 95% confidence intervals are tight (less than 0.005 in width) and are not shown.

We next compare SCBEBF against other alternatives. There are only a few methods that can be compared against our algorithm due to the high dimensional feature space. We compare against Compressed LSTD (CLSTD) [14], $L^2$-Regularized LSTD using a Biconjugate gradient solver (L2-LSTD), and $L^1$-Regularized LSTD using LARS-TD [2] with a Biconjugate gradient solver in the inner loop (L1-LSTD). These conjugate gradient solvers exploit the sparsity of the feature space to converge faster to the solution of linear equations [26]. We avoided online and stochastic gradient type methods as they are not very efficient in sample complexity.

We compare the described methods while increasing the size of the training set. The projection schedule for SCBEBF is set to $d = \lfloor 500e^{-i/300} \rfloor$ for all sample sizes. The regularization parameter of L2-LSTD was chosen among a small set of values using $1/5$ of the training data as validation set. Due to memory and time constraints, the optimal choice of parameters could not be set for CLSTD and L1-LSTD. The maximum size of projection for CLSTD and the maximum number of non-zero coefficients for L1-LSTD was set to 3000. CLSTD would run out of memory and L1-LSTD would take multiple hours if we increase these limits.

The results, averaged over 5 runs, are shown in Figure 1 (right). We see that L2-LSTD outperforms other methods, closely followed by SCBEBF. Not surprisingly, L1-LSTD and CLSTD are not competitive here as they are suboptimal with the mentioned constraints. This is a consequence of the fact that these algorithms scale worse with respect to memory and time complexity.

We conjecture that L2-LSTD is benefiting from the sparsity of the features space, not only in running time (due to the use of conjugate gradient solvers), but also in sample complexity. This makes L2-LSTD an attractive choice when the features are observed in the sparse basis. However, if the features are sparse in some unknown basis (observation is not sparse), then the time complexity of any linear solver in the observation basis can be prohibitive. SCBEBF, however, scales much better in such cases as the main computation is done in the compressed space.

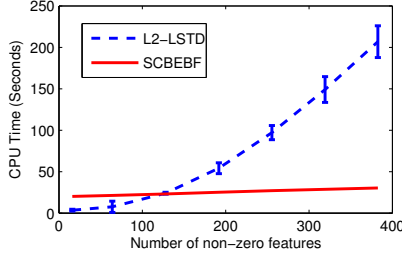

Figure 2: Runtime of L2-LSTD and SCBEBF with varying observation sparsity.

To highlight this effect, we construct an experiment in which we gradually increase the number of non-zero features using a change of basis. The error of both L2-LSTD and SCBEBF remain mostly unchanged as predicted by the theory. We thus only compare the running times as we change the observation sparsity. Figure 2 shows the CPU time used by each methods with sample size of 3000, averaged over 5 runs (using Matlab on a 3.2GHz Quad-Core Intel Xeon processor). We run 100 iterations of SCBEBF with $d = \lfloor 300e^{-i/30} \rfloor$ (as in the first experiment), and set the regularization parameter of L2-LSTD to the optimal value. We can see that the running time L2-LSTD quickly becomes prohibitive with the decreased observation sparsity, whereas the running time of SCBEBF grows very slowly (and linearly).

## 5   Discussion

We provided a simple, fast and robust feature extraction algorithm for policy evaluation in sparse and high dimensional state spaces. Using recent results on the properties of random projections, we proved that in sparse spaces, random projections of sizes logarithmic in the original dimension are sufficient to preserve linearity. Therefore, BEBFs can be generated on compressed spaces induced by small random projections. Our finite sample analysis provides guarantees on the reduction in prediction error after the addition of such BEBFs.

Our assumption of the linearity of the Bellman error in the original feature space might be too strong for some problems. We introduced this assumption to simplify the analysis. However, most of the discussion can be rephrased in terms of the *projected Bellman error*, and we expect this approach to carry through and provide more general results (e.g. see Parr et al. [6]).

Compared to other regularization approaches to RL [2, 27, 28], our random projection method does not require complex optimization, and thus is faster and more scalable. If features are observed in the sparse basis, then conjugate gradient solvers can be used for regularized value function approximation. However, CBEBF seems to have better performance with smaller sample sizes and provably works under any observation basis.

Finding the optimal choice of the projection size schedule and the number of iterations is an interesting subject of future research. We expect the use of cross-validation to suffice for the selection of the optimal parameters, due to the robustness that we observed in the results of the algorithm. A tighter theoretical bound might also help provide an analytical, closed-form answer to how parameters should be selected. One would expect a slow reduction in the projection size to be favourable.

**Acknowledgements:** Financial support for this work was provided by Natural Sciences and Engineering Research Council Canada, through their Discovery Grants Program.

## Footnotes

[1] For the more general case, the analysis can be done with respect to the *projected* Bellman error [6]. We assume linearity of the Bellman error to simplify the derivations.

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
