[Supplementary Material]

# Bellman Error Based Feature Generation using Random Projections on Sparse Spaces Appendix

**Mahdi Milani Fard, Yuri Grinberg, Amir massoud Farahmand, Joelle Pineau, Doina Precup**
School of Computer Science
McGill University
Montreal, Canada
{mmilan1,ygrinb,amirf,jpineau,dprecup}@cs.mcgill.ca

## 1  Preliminaries

To obtain a finite sample bound on the error of our algorithm, we require a mixing condition on the Markov chain induced by a given fixed policy in an MDP. Specifically we assume that the Markov chain *uniformly quickly forgets its past.* For such chains, we present here an extension of Bernstein's inequality based on Samson [1].

Let $\mathbf{x}_1, \ldots, \mathbf{x}_n$ be a time-homogeneous Markov chain with transition kernel $T(\cdot|\cdot)$ taking values in some measurable space $\mathcal{X}$. Consider the concentration of the average of the Markov Process:

$$(\mathbf{x}_1, f(\mathbf{x}_1)), \ldots, (\mathbf{x}_n, f(\mathbf{x}_n)), \tag{1}$$

where $f : \mathcal{X} \to [0, b]$ is a fixed measurable function. To arrive at a concentration inequality, we need a characterization of how fast $(\mathbf{x}_i)$ forgets its past.

Let $T^i(\cdot|x)$ be the $i$-step transition kernel: $T^i(A|x) = \Pr\{\mathbf{x}_{i+1} \in A \,|\, \mathbf{x}_1 = \mathbf{x}\}$ (for all $A \subset \mathcal{X}$ measurable). Define upper-triangular matrix $\mathbf{\Gamma}_n = (\gamma_{ij}) \in \mathbb{R}^{n \times n}$ as:

$$\gamma_{ij}^2 = \sup_{(\mathbf{x}, \mathbf{y}) \in \mathcal{X}^2} \|T^{j-i}(\cdot|\mathbf{x}) - T^{j-i}(\cdot|\mathbf{y})\|_{\mathrm{TV}}, \tag{2}$$

for $1 \le i < j \le n$ and let $\gamma_{ii} = 1$ $(1 \le i \le n)$. The operator norm of this matrix $\|\mathbf{\Gamma}_n\|$ w.r.t. the Euclidean distance, is a measure of dependence for the random sequence $\mathbf{x}_1, \mathbf{x}_2, \ldots, \mathbf{x}_n$. For example with independent $\mathbf{x}_i$'s, $\mathbf{\Gamma}_n = \mathbf{I}$ and $\|\mathbf{\Gamma}_n\| = 1$. In general $\|\mathbf{\Gamma}_n\|$, which appears in our concentration inequalities for dependent sequences, can grow with $n$. We can see that the "effective" sample size is $n/\|\mathbf{\Gamma}_n\|^2$.

We say that a time-homogeneous Markov chain *uniformly quickly forgets its past* if:

$$\tau = \sup_{n \ge 1} \|\mathbf{\Gamma}_n\|^2 < +\infty. \tag{3}$$

We refer to $\tau$ as the *forgetting time* of the chain. Conditions under which a Markov chain uniformly quickly forgets its past are of major interest. For further discussion on this, see [2].

The following result from [2] is a trivial corollary of Theorem 2 of [1]. Samson's result is stated for empirical processes and can be considered as a generalization of Talagrand's inequality to dependent random variables.

**Theorem 6** ([2]). *Let $f$ be a measurable function on $\mathcal{X}$ whose values lie in $[0, b]$, $(\mathbf{x}_i)_{1 \le i \le n}$ be a homogeneous Markov chain taking values in $\mathcal{X}$ with forgetting time $\tau$. Let $z = \frac{1}{n} \sum_{i=1}^n f(\mathbf{x}_i)$. For all $\epsilon \ge 0$:*

$$\mathbb{P}(z - \mathbb{E}[z] \ge \epsilon) \le \exp\left(-\frac{\epsilon^2 n}{2b\tau(\mathbb{E}[z] + \epsilon)}\right),$$

$$\mathbb{P}(\mathbb{E}[z] - z \ge \epsilon) \le \exp\left(-\frac{\epsilon^2 n}{2b\tau\mathbb{E}[z]}\right).$$

We use the above concentration theorem to provide a finite sample bound on the error of regression with non i.i.d. data.

## 2  Proof of Theorem 4

*Proof.* To begin the proof of the main theorem, first note that we can write the TD-errors as the sum of Bellman errors and some noise term: $\delta_t = e_V(\mathbf{x}_t) + \eta_t$. These noise terms form a series of martingale differences, as their expectation is 0 given all the history up to that point:

$$\mathbb{E}\left[\eta_t | \mathbf{x}_1 \dots \mathbf{x}_t, r_1 \dots r_{t-1}\right] = 0. \tag{4}$$

We also have that the Bellman error is linear in the features, thus in vector form:

$$\delta = \mathbf{X}\mathbf{w} + \eta. \tag{5}$$

Using random projections, in the compressed space we have:

$$\delta = (\mathbf{X}\mathbf{\Phi})(\mathbf{\Phi}^T\mathbf{w}) + \mathbf{b} + \eta, \tag{6}$$

where $\mathbf{b}$ is the vector of bias due to the projection. We have from Lemma 1 that with probability $1 - \xi/4$, for all $\mathbf{x} \in \mathcal{X}$:

$$
\begin{aligned}
\left|(\mathbf{x}^T\mathbf{\Phi})(\mathbf{\Phi}^T\mathbf{w}) - e_V(\mathbf{x})\right| &= \left|(\mathbf{x}^T\mathbf{\Phi})(\mathbf{\Phi}^T\mathbf{w}) - \mathbf{x}^T\mathbf{w}\right| \\
&\leq \epsilon_{\text{prj}}^{(\xi/4)}\|\mathbf{w}\|\|\mathbf{x}^T\|.
\end{aligned}
$$

Thus, $\mathbf{b}$ is element-wise bounded in absolute value by $\epsilon_{\text{prj}}^{(\xi/4)}\|\mathbf{w}\|$ with high probability. The weighted $L^2$ error in regression to the TD-error as compared to the Bellman error will be:

$$
\begin{aligned}
\left\|\mathbf{x}^T\mathbf{\Phi}\mathbf{w}_{\text{ols}}^{(\mathbf{\Phi})} - e_V(\mathbf{x})\right\|_\rho &= \left\|(\mathbf{x}^T\mathbf{\Phi})(\mathbf{X}\mathbf{\Phi})^\dagger[(\mathbf{X}\mathbf{\Phi})(\mathbf{\Phi}^T\mathbf{w}) + \mathbf{b} + \eta] - e_V(\mathbf{x})\right\|_\rho \\
&= \left\|(\mathbf{x}^T\mathbf{\Phi})(\mathbf{\Phi}^T\mathbf{w}) - e_V(\mathbf{x}) + (\mathbf{x}^T\mathbf{\Phi})(\mathbf{X}\mathbf{\Phi})^\dagger\mathbf{b} + (\mathbf{x}^T\mathbf{\Phi})(\mathbf{X}\mathbf{\Phi})^\dagger\eta\right\|_\rho \\
&\leq \left\|(\mathbf{x}^T\mathbf{\Phi})(\mathbf{\Phi}^T\mathbf{w}) - e_V(\mathbf{x})\right\|_\rho + \left\|(\mathbf{x}^T\mathbf{\Phi})(\mathbf{X}\mathbf{\Phi})^\dagger\mathbf{b}\right\|_\rho + \left\|(\mathbf{x}^T\mathbf{\Phi})(\mathbf{X}\mathbf{\Phi})^\dagger\eta\right\|_\rho \\
&\leq \epsilon_{\text{prj}}^{(\xi/4)}\|\mathbf{w}\|\|\mathbf{x}\|_\rho + \left\|(\mathbf{x}^T\mathbf{\Phi})(\mathbf{X}\mathbf{\Phi})^\dagger\mathbf{b}\right\|_\rho + \left\|(\mathbf{x}^T\mathbf{\Phi})(\mathbf{X}\mathbf{\Phi})^\dagger\eta\right\|_\rho.
\end{aligned}
$$

The second term is the regression to the bias, and the third term is the regression to the noise. We present lemmas that bound these terms. The theorem is proved by the application and union bounding of of Lemmas 9, 11 and 1 with $\xi_0 = \xi/4$. $\square$

### 2.1  Bounding the Regression to Bias Terms

To bound the regression to the bias term, we need the following concentration lemmas based on Theorem 6 for fast mixing Markov chains:

**Lemma 7.** *Under the conditions of Theorem 6, for any $0 < \xi < 1$, w.p. $1 - \xi$:*

$$z \leq 2\,\mathbb{E}\left[z\right] + \frac{4b\tau}{n}\,\log\frac{1}{\xi}. \tag{7}$$

*Proof.* Since $\mathbb{E}\left[z\right] \geq 0$, using Theorem 6 we have for any $\epsilon > 0$:

$$
\begin{aligned}
\mathbb{P}\left(z - 2\,\mathbb{E}\left[z\right] \geq \epsilon\right) &= \mathbb{P}\left(z - \mathbb{E}\left[z\right] \geq \mathbb{E}\left[z\right] + \epsilon\right) & (8) \\
&\leq \exp\left(-\frac{(\mathbb{E}\left[z\right] + \epsilon)^2\,n}{2b\tau(2\,\mathbb{E}\left[z\right] + \epsilon)}\right) & (9) \\
&\leq \exp\left(-\frac{(\mathbb{E}\left[z\right] + \epsilon)\,n}{4b\tau}\right) & (10) \\
&\leq \exp\left(-\frac{\epsilon\,n}{4b\tau}\right). & (11)
\end{aligned}
$$

The lemma follows by solving for $\epsilon$. $\square$

**Lemma 8.** *Under the conditions of Theorem 6, for any $0 < \xi < 1$, w.p. $1 - \xi$:*

$$\mathbb{E}\left[z\right] \leq 2z + \frac{8b\tau}{n} \log \frac{1}{\xi}. \tag{12}$$

*Proof.* Since $\mathbb{E}\left[z\right] \geq 0$, using Theorem 6 we have for any $\epsilon > 0$:

$$\mathbb{P}\left(\mathbb{E}\left[z\right] - 2z \geq \epsilon\right) = \mathbb{P}\left(\mathbb{E}\left[z\right] - z \geq (\mathbb{E}\left[z\right] + \epsilon)/2\right) \tag{13}$$

$$\leq \exp\left(-\frac{(\mathbb{E}\left[z\right] + \epsilon)^2 n}{8b\tau\mathbb{E}\left[z\right]}\right) \tag{14}$$

$$\leq \exp\left(-\frac{(\mathbb{E}\left[z\right] + \epsilon) n}{8b\tau}\right) \tag{15}$$

$$\leq \exp\left(-\frac{\epsilon n}{8b\tau}\right) \tag{16}$$

$$\tag{17}$$

The lemma follows by solving for $\epsilon$. $\square$

**Lemma 9** (Bounding regression to the bias)**.** *Under the conditions of Theorem 4 and assuming inner products are preserved in Lemma 1 with $\epsilon_{prj}^{(\xi/4)}$, with probability no less than $1 - \xi/2$:*

$$\left\|(\mathbf{x}^T\boldsymbol{\Phi})\mathbf{w}_{\mathbf{X}}\right\|_\rho \leq 11\alpha\epsilon_{prj}^{(\xi/4)}\|\mathbf{w}\|\|\mathbf{x}\|_\rho\sqrt{\frac{1}{d\nu}} + 4\alpha\epsilon_{prj}^{(\xi/4)}\|\mathbf{w}\|\sqrt{\frac{d\tau}{n\nu}\log\frac{d}{\xi_1}}. \tag{18}$$

*Proof.* Define $\mathbf{w}_{\mathbf{X}} = (\mathbf{X}\boldsymbol{\Phi})^\dagger\mathbf{b}$. Also define $\|.\|_n$ to be the weighted $L^2$ norm uniform on the sample set $X$:

$$\|f(\mathbf{x})\|_n^2 = \frac{1}{n}\sum_{i=1}^{n}(f(\mathbf{X}_i))^2. \tag{19}$$

We start by bounding the empirical norm $\|(\mathbf{x}^T\boldsymbol{\Phi})\mathbf{w}_{\mathbf{X}}\|_n$. Given that $(\mathbf{X}\boldsymbol{\Phi})\mathbf{w}_{\mathbf{X}}$ is the OLS regression to the bias on the observed points, its sum of squared errors should not be greater than any other linear regression, including the vector 0, thus $\|(\mathbf{x}^T\boldsymbol{\Phi})\mathbf{w}_{\mathbf{X}} - b(\mathbf{x})\|_n \leq \|b(\mathbf{x})\|_n$. We get:

$$\|(\mathbf{x}^T\boldsymbol{\Phi})\mathbf{w}_{\mathbf{X}}\|_n \leq \|(\mathbf{x}^T\boldsymbol{\Phi})\mathbf{w}_{\mathbf{X}} - b(\mathbf{x})\|_n + \|b(\mathbf{x})\|_n$$
$$\leq 2\|b(\mathbf{x})\|_n \leq 2\epsilon_{prj}^{(\xi/4)}\|\mathbf{w}\|\|\mathbf{x}\|_n. \tag{20}$$

Let $\mathcal{W} = \{\mathbf{u} \in \mathbb{R}^d \text{ s.t. } \|\mathbf{u}\| \leq 1\}$. Let $S \subset \mathcal{W}$ be an $\epsilon$-grid cover of $\mathcal{W}$:

$$\forall \mathbf{v} \in \mathcal{W} \, \exists \mathbf{u} \in S : \|\mathbf{u} - \mathbf{v}\| \leq \epsilon. \tag{21}$$

It is easy to prove (see e.g. Chapter 13 of [3]) that these conditions can be satisfied by choosing a grid of size $|S| \leq (3/\epsilon)^d$ ($S$ fills up the space within $\epsilon$ distance). Applying union bound to Lemma 8 (let $f(\mathbf{x}) = ((\mathbf{x}^T\boldsymbol{\Phi})\mathbf{u})^2$) for all elements in $S$, we get with probability no less than $1 - \xi/4$, for all $\mathbf{u} \in S$:

$$\|(\mathbf{x}^T\boldsymbol{\Phi})\mathbf{u}\|_\rho^2 \leq 2\|(\mathbf{x}^T\boldsymbol{\Phi})\mathbf{u}\|_n^2 + \frac{8\alpha^2\tau}{n}\log\frac{4|S|}{\xi}, \tag{22}$$

which yields the following after simplification:

$$\|(\mathbf{x}^T\boldsymbol{\Phi})\mathbf{u}\|_\rho \leq \sqrt{2}\|(\mathbf{x}^T\boldsymbol{\Phi})\mathbf{u}\|_n + \alpha\sqrt{\frac{8\tau}{n}\log\frac{4|S|}{\xi}}. \tag{23}$$

Let $\mathbf{w}'_{\mathbf{X}} = \mathbf{w}_{\mathbf{X}} / \|\mathbf{w}_{\mathbf{X}}\|$. For any $\mathbf{X}$, since $\mathbf{w}'_{\mathbf{X}} \in \mathcal{W}$, there exists $\mathbf{w}'' \in S$ such that $\|\mathbf{w}'_{\mathbf{X}} - \mathbf{w}''\| \leq \epsilon$. Therefore, under event (23) we have:

$$
\begin{align}
\left\|(\mathbf{x}^T\boldsymbol{\Phi})\mathbf{w}_{\mathbf{X}}\right\|_\rho / \|\mathbf{w}_{\mathbf{X}}\| &= \left\|(\mathbf{x}^T\boldsymbol{\Phi})\mathbf{w}'_{\mathbf{X}}\right\|_\rho \tag{24}\\
&\leq \left\|(\mathbf{x}^T\boldsymbol{\Phi})(\mathbf{w}'_{\mathbf{X}} - \mathbf{w}'')\right\|_\rho + \left\|(\mathbf{x}^T\boldsymbol{\Phi})\mathbf{w}''\right\|_\rho \tag{25}\\
&\leq \left\|\mathbf{x}^T\boldsymbol{\Phi}\right\|_\rho \|\mathbf{w}'_{\mathbf{X}} - \mathbf{w}''\| + \sqrt{2}\left\|(\mathbf{x}^T\boldsymbol{\Phi})\mathbf{w}''\right\|_n \notag\\
&\quad + \alpha\sqrt{(8\tau/n)\log(4|S|/\xi)} \tag{26}\\
&\leq \alpha\|\mathbf{x}\|_\rho\epsilon + \sqrt{2}\left\|(\mathbf{x}^T\boldsymbol{\Phi})(\mathbf{w}'' - \mathbf{w}'_{\mathbf{X}})\right\|_n + \sqrt{2}\left\|(\mathbf{x}^T\boldsymbol{\Phi})\mathbf{w}'_{\mathbf{X}}\right\|_n \notag\\
&\quad + \alpha\sqrt{(8\tau/n)\log(4|S|/\xi)} \tag{27}\\
&\leq \alpha\|\mathbf{x}\|_\rho\epsilon + \sqrt{2}\alpha\|\mathbf{x}\|_n\epsilon + \sqrt{2}\left\|(\mathbf{x}^T\boldsymbol{\Phi})\mathbf{w}'_{\mathbf{X}}\right\|_n \notag\\
&\quad + \alpha\sqrt{(8\tau/n)\log(4|S|/\xi)} \tag{28}\\
&\leq \sqrt{2}\left\|(\mathbf{x}^T\boldsymbol{\Phi})\mathbf{w}_{\mathbf{X}}\right\|_n / \|\mathbf{w}_{\mathbf{X}}\| + \alpha\epsilon(\|\mathbf{x}\|_\rho + \sqrt{2}\|\mathbf{x}\|_n) \notag\\
&\quad + \alpha\sqrt{(8\tau/n)\log(4|S|/\xi)}. \tag{29}
\end{align}
$$

Line (26) uses Equation (23), and we use Equation (21) in lines (27) and (28). Using the definition, we have that $\|\mathbf{w}_{\mathbf{X}}\| \leq \left\|(\mathbf{X}\boldsymbol{\Phi})^\dagger\right\| \epsilon_{\mathrm{prj}}^{(\xi/4)}\|\mathbf{w}\|\sqrt{n} \leq \epsilon_{\mathrm{prj}}^{(\xi/4)}\|\mathbf{w}\|\sqrt{1/\nu}$. Thus, using Equation (20) we get:

$$
\begin{align}
\left\|(\mathbf{x}^T\boldsymbol{\Phi})\mathbf{w}_{\mathbf{X}}\right\|_\rho &\leq \sqrt{8}\epsilon_{\mathrm{prj}}^{(\xi/4)}\|\mathbf{w}\|\|\mathbf{x}\|_n + \alpha\epsilon_{\mathrm{prj}}^{(\xi/4)}\|\mathbf{w}\|\epsilon\sqrt{1/\nu}\left(\|\mathbf{x}\|_\rho + \sqrt{2}\|\mathbf{x}\|_n\right) \notag\\
&\quad + \alpha\epsilon_{\mathrm{prj}}^{(\xi/4)}\|\mathbf{w}\|\sqrt{\frac{8\tau}{n\nu}\log\frac{4|S|}{\xi}}. \tag{30}
\end{align}
$$

Using Lemma 7 on the squared norm of $\mathbf{x}$, we get with probability no less than $1 - \xi/4$:

$$
\|\mathbf{x}\|_n^2 \leq 2\|\mathbf{x}\|_\rho^2 + \frac{4\tau}{n}\log\frac{4}{\xi}, \tag{31}
$$

which yields the following after simplification:

$$
\|\mathbf{x}\|_n \leq \sqrt{2}\|\mathbf{x}\|_\rho + 2\sqrt{\frac{\tau}{n}\log\frac{4}{\xi}}. \tag{32}
$$

Setting $\epsilon = 1/\sqrt{d}$, using Equation (32) and substituting $|S|$ into (30) we get:

$$
\begin{align}
\left\|(\mathbf{x}^T\boldsymbol{\Phi})\mathbf{w}_{\mathbf{X}}\right\|_\rho &\leq (8 + 3\alpha\sqrt{1/d\nu})\epsilon_{\mathrm{prj}}^{(\xi/4)}\|\mathbf{w}\|\|\mathbf{x}\|_\rho \notag\\
&\quad + (\sqrt{32} + \alpha\sqrt{8/d\nu})\epsilon_{\mathrm{prj}}^{(\xi/4)}\|\mathbf{w}\|\sqrt{\frac{\tau}{n}\log\frac{4}{\xi}} \notag\\
&\quad + \alpha\epsilon_{\mathrm{prj}}^{(\xi/4)}\|\mathbf{w}\|\sqrt{\frac{8\tau}{n\nu}\log\frac{4(3\sqrt{d})^d}{\xi}}. \tag{33}
\end{align}
$$

Since $d \geq 10$, $\nu \leq \alpha^2/d$ and $\alpha \geq 1$ we have:

$$
\begin{align}
\left\|(\mathbf{x}^T\boldsymbol{\Phi})\mathbf{w}_{\mathbf{X}}\right\|_\rho &\leq 11\alpha\sqrt{\frac{1}{d\nu}}\epsilon_{\mathrm{prj}}^{(\xi/4)}\|\mathbf{w}\|\|\mathbf{x}\|_\rho \notag\\
&\quad + 9\,\alpha\sqrt{\frac{1}{d\nu}}\epsilon_{\mathrm{prj}}^{(\xi/4)}\|\mathbf{w}\|\sqrt{\frac{\tau}{n}\log\frac{4}{\xi}} \notag\\
&\quad + 3\alpha\epsilon_{\mathrm{prj}}^{(\xi/4)}\|\mathbf{w}\|\sqrt{\frac{d\tau}{n\nu}\log\frac{d}{\xi}}. \tag{34}
\end{align}
$$

Union bounding over the events of Eqn (22) and (31) gives the lemma after simplification. $\qquad\square$

## 2.2 Bounding the Regression to Noise Terms

To bound the regression to the noise, we need the following lemma on martingales:

**Lemma 10.** *Let* $\mathbf{y}$ *be a vector of size* $n \times 1$, *in which row* $t$ *is a function of* $\mathbf{x}_t$. *Then with probability* $1 - \xi$ *we have:*

$$|\mathbf{y}^T \eta| \leq \delta_{\max} \|\mathbf{y}\| \sqrt{2 \log \frac{2}{\xi}}. \tag{35}$$

*Proof.* This is a simple application of a concentration inequality on martingales. $\qquad \square$

**Lemma 11** (Bounding regression to the noise). *Under the conditions of Theorem 4 and assuming inner products are preserved in Lemma 1 with* $\epsilon_{prj}^{(\xi/4)}$, *with probability no less than* $1 - \xi/4$:

$$\left\| (\mathbf{x}^T \mathbf{\Phi})(\mathbf{X}\mathbf{\Phi})^\dagger \eta \right\|_\rho \leq 2\alpha \delta_{\max} \|\mathbf{x}\|_\rho \sqrt{\frac{\kappa d}{n\nu} \log \frac{d}{\xi}}. \tag{36}$$

*Proof.* For all $i \in \{1, \ldots, d\}$, define the vector $\mathbf{e}_i^{d \times 1}$ to have 1 on the $i$th row and be 0 elsewhere. Using union bound on Lemma 10, we have with probability no less than $1 - \xi/4$:

$$\forall i : \left| e_i^T (\mathbf{X}\mathbf{\Phi})^T \eta \right| \leq \delta_{\max} \|\mathbf{X}\mathbf{\Phi}\| \sqrt{2 \log \frac{8d}{\xi}}. \tag{37}$$

For any fixed $\mathbf{x} \in \mathcal{X}$, define $\mathbf{y}^T = (\mathbf{x}^T \mathbf{\Phi})((\mathbf{X}\mathbf{\Phi})^T(\mathbf{X}\mathbf{\Phi}))^{-1}$. We have:

$$\left| (\mathbf{x}^T \mathbf{\Phi})(\mathbf{X}\mathbf{\Phi})^\dagger \eta \right| = \left| \mathbf{y}^T (\mathbf{X}\mathbf{\Phi})^T \eta \right| \tag{38}$$

$$= \left| \sum_{i=1}^d (\mathbf{y}^T \mathbf{e}_i) \mathbf{e}_i^T (\mathbf{X}\mathbf{\Phi})^T \eta \right| \tag{39}$$

$$\leq \sum_{i=1}^d \left| \mathbf{y}^T \mathbf{e}_i \right| \left| \mathbf{e}_i^T (\mathbf{X}\mathbf{\Phi})^T \eta \right| \tag{40}$$

$$\leq \delta_{\max} \|\mathbf{X}\mathbf{\Phi}\| \sqrt{2 \log \frac{8d}{\xi}} \|\mathbf{y}\|_1 \tag{41}$$

$$\leq \delta_{\max} \|\mathbf{X}\mathbf{\Phi}\| \sqrt{2d \log \frac{8d}{\xi}} \|\mathbf{y}\|. \tag{42}$$

Therefore we get:

$$\left\| (\mathbf{x}^T \mathbf{\Phi})(\mathbf{X}\mathbf{\Phi})^\dagger \eta \right\|_\rho \leq \delta_{\max} \|\mathbf{X}\mathbf{\Phi}\| \left\| (\mathbf{x}^T \mathbf{\Phi})((\mathbf{X}\mathbf{\Phi})^T(\mathbf{X}\mathbf{\Phi}))^{-1} \right\|_\rho \sqrt{2d \log \frac{8d}{\xi}} \tag{43}$$

$$\leq \alpha \delta_{\max} \|\mathbf{x}\|_\rho \|\mathbf{X}\mathbf{\Phi}\| \left\| ((\mathbf{X}\mathbf{\Phi})^T(\mathbf{X}\mathbf{\Phi}))^{-1} \right\| \sqrt{2d \log \frac{8d}{\xi}} \tag{44}$$

$$\leq \alpha \delta_{\max} \|\mathbf{x}\|_\rho \sqrt{\frac{2\kappa d}{n\nu} \log \frac{8d}{\xi}}, \tag{45}$$

which gives the lemma after simplification. $\qquad \square$

# 3 Proof of Lemma 3

*Proof.* We have that $V^\pi$ is the fixed point to the Bellman operator (i.e. $\mathcal{T}V^\pi = V^\pi$), and that the operator is a contraction with respect to the weighted $L^2$ norm on the stationary distribution $\rho$ [4]:

$$\|\mathcal{T}V(\mathbf{x}) - \mathcal{T}V'(\mathbf{x})\|_\rho \leq \gamma \|V(\mathbf{x}) - V'(\mathbf{x})\|_\rho. \tag{46}$$

We thus have:

$$
\begin{aligned}
\|V^\pi(\mathbf{x}) - (V(\mathbf{x}) + \psi(\mathbf{x}))\|_\rho &\leq \|V^\pi(\mathbf{x}) - \mathcal{T}V(\mathbf{x})\|_\rho + \|(\mathcal{T}V(\mathbf{x}) - V(\mathbf{x})) - \psi(\mathbf{x})\|_\rho & (47)\\
&\leq \|\mathcal{T}V^\pi(\mathbf{x}) - \mathcal{T}V(\mathbf{x})\|_\rho + \epsilon\|\mathcal{T}V(\mathbf{x}) - V(\mathbf{x})\|_\rho & (48)\\
&\leq \gamma\|V^\pi(\mathbf{x}) - V(\mathbf{x})\|_\rho \\
&\quad + \epsilon\|\mathcal{T}V(\mathbf{x}) - \mathcal{T}V^\pi(\mathbf{x})\|_\rho + \epsilon\|V^\pi(\mathbf{x}) - V(\mathbf{x})\|_\rho & (49)\\
&\leq (\gamma + \epsilon\gamma + \epsilon)\|V^\pi(\mathbf{x}) - V(\mathbf{x})\|_\rho. & (50)
\end{aligned}
$$

$\square$

# 4 Proof of Lemma 4

*Proof.* Let $\epsilon = (\gamma_0 - \gamma)/(1 + \gamma)$, $c_4 = 25600$ and $c_5 = 64$. Substituting $d$, and $n$ into Theorem 2, after simplification, with probability $1 - \xi$ we get: $\|\mathbf{x}^T\mathbf{\Phi}\mathbf{w}_{\text{ols}}^{(\Phi)} - \mathbf{x}^T\mathbf{w}\|_{\rho(\mathbf{x})} \leq \epsilon\|\mathbf{x}^T\mathbf{w}\|_\rho$. Proof follows immediately by an application of Lemma 3. $\square$

# 5 CBEBF With Compressed Ridge Regression

The dependence of the bound of Theorem 2 on the smallest eigenvalue of the empirical gram matrix can be linked to the properties of the pseudo inverse and its use in OLS regression. To avoid such dependence, we might need to use an extra level of regularization in the compressed space.

One possible solution is the use of ridge regression instead of OLS in the inner loop of our algorithm. The detailed analysis of the error rate of such algorithm is beyond the scope of this work, but we expect the dependence on $\nu$ to be replaced by the regularization factor of the ridge regression, denoted by $\lambda$, with the addition of an extra bias factor. An optimal choice for $\lambda$ can be found either using an upper bound on the error rate, or empirically using cross validation.