[Reviews · NeurIPS 2013]

Submitted by Assigned_Reviewer_2

The construction of features for use in value function approximation for reinforcement learning is a very active area of research. In this paper, the authors develop and test an approach based on random projections. A proof of convergence is provided.

In Section 1, paragraphs 2 and 3, please explain how "computationally expensive" the approaches, such as [6], are. The claim of superior efficient for the method proposed here would be better supported of the approach of [6] were included in the experiments in Section 4.

In Section 3, it would be helpful to include some of the justification, from [6], for the statement that the Bellman error is linearly representable in the original state feature space.

In Section 4, second paragraph, the tile-coding for the helicopter state space is described. It is said that it is used "for demonstration purposes". Here it would be helpful to discuss alternative representations, and why or why not they may not work. Perhaps you use such a high-dimensional (65,536) feature space to increase the likelihood that the Bellman error is linear in that space.

In Section 4, fifth paragraph, you say that there are few methods that can be compared to yours, because of the high-dimensional feature space. But, the original state only contains 12 continuous variables; many function approximation methods can be applied directly to that space. This must be mentioned, and, ideally, your experiments expanded to include comparisons to more straightforward methods that operate directly on the continuous state space.
Summary: Recent results in random projections have been adopted for the reinforcement learning paradigm. In this paper the authors take the novel approach of developing a method based on random projections to incrementally construct new features based on Bellman errors. A convergence proof is provided. Empirical evidence shows the new approach achieves comparable accuracy at a much reduced computation time.

Submitted by Assigned_Reviewer_4

This paper proposes to use random projections as a proxy to learn BEBFs (Bellman Error Basis Functions). Given a (high dimensional) set of features and the currently estimated value function, the features are (randomly) projected on a smaller space, and the temporal differences errors (related to the currently estimated value function) are regressed on these projected features. The (scalar) regressed function is then added to the set of features used to estimate the value function. A finite sample analysis is conducted, the main result showing that if the Bellman residual is linear in the (high dimensional) features, then the Bellman error can be well regressed on the compressed space (depending notably on the size of this space and on the number of samples). The authors also use this result to provide some guarantee on the estimated value function. The proposed algorithm is compared to state-of-the-art approaches on a high dimensional problem.

This paper is very clearly written and presents some quite interesting contributions, summed up above. I have mainly two comments.
1) First, I found that lemma 4 was not so clear, which is quite a pity as it summarizes the preceding results and quantifies the efficiency of the proposed approach. Notably, how strong is the assumption of the existence of constants c1 to c3? Also, is it straightforward that the universal (?) constants c4 and c5 exist (or is this an assumption?). Considering the provided bound, we should get $\gamma_0$ as small as possible, which seems actually a good thing also for the choice of $d$ and $n$... this seems quite surprising. What did I miss? I think that a proof (in the appendix) may help (even if considered as straightforward)
2) Second, I have some comments regarding the experimental section:
* what is the reward?
* it would be interesting to consider also CBEBF (the not simple version) in the experiments, even if it has a higher computational cost, to see what can be gained in estimation quality (above l2-LSTD?)
* the results of l2-LSTD are quite surprising... a validation set is mentioned, but cross-validation for value estimation does not seem to be straightforward. Do you use the NRPE to cross-validate?
* it would have been interesting to compare your algorithm to iLSTD [1], which pursue a similar goal (sample efficiency and low computational cost)
* why do you change the schedule for dimensions between experiments (as it is the same problem)?
* what is the $\mu$ distribution in the experiment (used for NRPE)?
* that is true that the gradient-based approaches should be less sample-efficient (and implies the choice of some possibly sensitive meta-parameters), but it would have been intersting to show it (empirically)
* regarding the last experiment, it would have been interesting to see if the efficiency of l2-LSTD degrades when the feature are not sparse (e.g. features transformed through a linear mapping). This is actually mentioned in the text (no change), but fig.2 suggests that the feature vector is quite sparse (400 non-zero over 65536).

[1] Alborz Geramifard, Michael Bowling, and Richard S. Sutton. Incremental Least-Squares Temporal Difference Learning. In AAAI. 2006
Summary: This paper introduces an algorithm which uses random projection as a proxy to learn BEBFs for value function estimation (with a highlighted computational gain compared to other approaches). The last part of the analysis could be clarified and the experiments could be a little bit improved, but this is a solid and interesting contribution.

Submitted by Meta_Reviewer_10

Paper attempts to make Bellman error based features (BEBF) more scalable and computational efficient by using the idea of random projections in very high dimensional sparse feature spaces. Finite time guarantees and empirical results are provided.
Summary: Considers a problem of interest and makes a clear advance in the field.
Author Feedback

Author rebuttal: We thank the reviewers for their useful comments.

Assigned_Reviewer_2:

- We used tile coding in our experiments as it is very commonly used in practice in the RL literature. Linear function approximation in the original space was not as good as tile coding (we get slightly better results by mixing the original features and tile-coded features). We expect higher order polynomial regression to work decently well (since the model represents a physical system), but such method is problem specific and not widely applicable. We thus chose this particular feature space for demonstration purposes and avoided further analysis on alternative feature spaces. We will clarify the text to reflect this point.


Assigned_Reviewer_4:

- Constants c4 and c5 exist (not an assumption).

- Decreasing gamma_0 requires us to use more samples and bigger projection sizes, and results in a better contraction of error. The lower bounds in the conditions of the lemma on "n" and "d" increase with smaller gamma_0.

- We did not find a good way to adapt iLSTD to our setting. Adding single features does not help decrease the error by much (as indicated by the LARS-TD results). Candidates features of iLSTD should be chosen in some other way (maybe among projections?).

- NRPE for GTD was off the error chart. We suspect we can tweak it with a better selection of parameters, but do not expect it to be a competitive alternative in a batch setting.

- CBEBF provided similar results to SCBEBF (slightly better when ridge regression was used for value function approximation). We decided to exclude the results to focus more on the comparison with other methods.

- The difference in the choice of projection schedule is due to the difference in sample sizes of each experiment. Note that the projection size should be chosen accordingly to optimize the error contraction (see lemma 4).

- The reward model is described in [25] (distance to a stable state at the origin) and the distribution of the sample is defined by the policy provided with their implementation (in RL-Glue).